∂ | Open Peer Review | Antimicrobial Chemotherapy | Research Article

# Comparative reassessment of AcrB efflux inhibitors reveals differential impact of specific pump mutations on the activity of potent compounds

Sabine Schuster,[1] Martina Vavra,[1] Dave A. N. Wirth,[1] Winfried V. Kern[1,2]

ABSTRACT   Multidrug resistance poses global challenges, particularly with regard to Gram-negative bacterial infections. In view of the lack of new antibiotics, drug enhancers, such as efflux pump inhibitors (EPIs), have increasingly come into focus. A number of chemically diverse agents have been reported to inhibit AcrB, the main multidrug transporter in *Escherichia coli*, and homologs in other Gram-negative bacteria. However, due to the often varying methodologies used for their characterization, results remain difficult to compare. In this study, using a defined selection of antibiotics known to be efflux substrates, we reevaluated 38 published compounds for their *in vitro* EPI activity. When examined in an *E. coli* strain with stable wild-type AcrB overexpression, we found 17 compounds showing at least fourfold enhancing potency with more than 2 out of 10 test drugs (belonging to eight antibiotic classes). Pyranopyridines (MBX series) were confirmed as the most potent inhibitors among agents reported so far. A new and surprising finding was that their activity, unlike that of the pyridylpiperazine EPI BDM88855, was highly susceptible to the AcrB double-mutation G141D_N282Y, which had previously been shown to diminish drug enhancing of 1-(1-naphthylmethyl)piperazine in a predominantly substrate-specific manner. Conversely, transmembrane region mutation V411A, while eliminating the drug potentiating of the BDM compound, did not decrease the activity of the MBX EPIs. Besides comparative reassessment of the potency of reported EPIs, the study demonstrated the usefulness of mutagenesis approaches providing tools for an initial discrimination of EPIs regarding their mode of function.

IMPORTANCE Infections with difficult-to-treat multidrug-resistant bacteria pose an urgent global threat in view of the stagnating development of new antimicrobial substances. Efflux pumps in Gram-negative pathogens are known to substantially contribute to multidrug resistance making them promising targets for chemotherapeutic interventions to restore the efficacy of conventional antibiotics. In the present study, the *in vitro* activity of previously reported efflux pump inhibitors was reassessed using standardized conditions. Relevant drug sensitizing activity could be proven for almost half of the tested compounds. Further characterization of potent inhibitors was achieved by investigating the impact of specific efflux pump mutations. A double-mutation previously known to decrease the activity of the arylpiperazine 1-(1-naphthylmethyl)piperazine also impaired that of the highly efficient pyranopyridine efflux pump inhibitors. Our findings provide direct comparability of reported efflux pump inhibitors and contribute to the elucidation of their mode of action.

KEYWORDS    efflux pump inhibitor, AcrB, RND-type transporter, multidrug resistance

Address correspondence to Sabine Schuster, sabine.schuster@uniklinik-freiburg.de.

The authors declare no conflict of interest.

See the funding table on p. 12.

Efflux by the resistance-nodulation-cell division (RND) superfamily transporters has been shown to make a non-negligible contribution to the development of multidrug

resistance (MDR) in Gram-negative pathogens (1, 2). In addition, they have been reported to play a role in biofilm formation (3), and for virulence and pathogenesis (4), making them promising targets for antimicrobial chemotherapy. RND transporters appear ubiquitous in the kingdom of life (5). In Gram-negative bacteria, they are typically embedded in a protein complex due to the requirement to overcome an inner and an outer membrane (OM). In *Escherichia coli*, the trimeric efflux pump AcrB is complexed with the OM channel TolC and membrane fusion proteins (AcrA) (6, 7). While further RND transporters are encoded in the *E. coli* genome, only AcrAB-TolC reveals constitutive expression in wild-type strains. Increased AcrB expression easily emerges under drug selection pressure (8). The compound extrusion procedure is proton-motive force driven (9) and triggered by cyclic conformational changes of the three AcrB protomers (10). MDR RND transporters contain a phenylalanine-rich distal and a large proximal substrate-binding pocket (DBP and PBP) (11, 12), which are separated by a glycine-rich flexible loop (designated as F617-, switch-, or G-loop) (12, 13). Because the substrate range of these efflux pumps is extremely broad and comprises almost all antibiotic classes and other noxious compounds (14, 15), exposure to a single drug potentially can cause MDR. This circumstance makes efflux by RND pumps an important target in the fight against Gram-negative pathogens.

The era of efflux pump inhibitor (EPI) research started in 1999 with the discovery of phenylalanine arginine β-naphthylamide (16), the first compound with demonstrated drug-enhancing potency due to an impact on Gram-negative RND transporters. Since then, a number of substances from various chemical classes have been reported to hamper such MDR efflux pumps. Among them are compounds from nature (17–23) and drugs developed for purposes other than antimicrobial therapy (24–31). Most EPIs have been derived from screenings of synthetic or half-synthetic compound libraries (16, 32–47) and were described to affect the well-studied multidrug RND transporter AcrB from *E. coli* and/or their close homologs in *Pseudomonas aeruginosa*, *Salmonella*, *Klebsiella*, and *Acinetobacter* species when using at sub-growth-inhibitory concentrations. Usually, their potency was evaluated by synergy testing in drug susceptibility assays, often in combination with intracellular dye accumulation studies (24, 27, 32, 36, 38, 48, 49). In addition, molecular dynamics and virtual docking studies (28, 47, 50), that have also been used for compound screening (17), and other methods such as co-crystallization with efflux proteins (47, 50, 51), cryo-EM (cryogenic electron microscopy) examinations (45), and mutagenesis approaches (47, 52, 53) provided indications regarding the mechanism of action. Despite increasing research advances and a remarkable number of agents published in the last two decades, no clinically applicable EPIs have been made available to date. However, well-studied model compounds with proven *in vitro* activity have served as powerful tools for the elucidation of drug transport pathways and critical sites regarding the inhibition of efflux in AcrB.

The comparability of reported EPI activity and potency remains difficult, even if they were studied with similar methods. Limitations include variable characterization procedures ranging from the use of only a single method to the application of several approaches. Comparison has also been difficult due to varying test strains, combinational drugs, and culture conditions. In the present study, we reevaluated published compounds with reported EPI activity in *E. coli* using a set of drugs from different antibiotic classes known to be efflux substrates and a test strain with stable overexpression of AcrB. Agents revealing sufficient activity according to our criteria were further examined regarding the impact of AcrB mutations previously shown to affect the activity of the EPI 1-(1-naphthylmethyl)piperazine (NMP) (52) or that of the recently reported new pyridylpiperazine EPI BDM88855 (47).

## RESULTS AND DISCUSSION

### Selected compounds

A literature search including publications by the end of 2022 yielded 38 compounds (Table 1) that (i) were described as inhibitors of AcrB in *E. coli*, and (ii) could be made available to us from a commercial supplier or directly from the research group. Notably, we were unable to obtain EPIs of the following substance groups: pyridopyrimidines (D13-9001) (34), 2-naphthamide derivatives [4-(isopentyloxy)-2-naphthamide] (39), 4-substituted quinazoline-2-carboxamide derivatives (41), 2H-benzo[h]chromene derivatives (44), and nordihydroguaretic acid analogues (40).

### Assessing the drug-enhancing potency

The obtained collection of compounds was subjected to combinational susceptibility testing according to a standard procedure. To achieve the most differential readout in terms of their potentiating activity, an *E. coli* strain with stable overexpression of the AcrB efflux pump was used as the test strain. The assays were carried out in combination with 10 drugs from eight different antibiotic classes all of them being confirmed substrates of AcrB. Under these conditions, 17 compounds revealed at least fourfold potentiating activity with more than two drugs. Broadest efficacies with best dose-effect relationship were confirmed for three of the pyranopyridine EPIs, MBX3132, MBX3135, and MBX3796, while broad, but somewhat more limited activity (higher doses needed) was found with BDM88855 (Table 1).

Notably, 18 out of 38 agents did not show any significant sensitizing with the drugs from our test series (Table 1). Discrepancies with previous findings may be due to the usage of different test antibiotics, conditions, and strains. In some cases, the activity had only been assumed because of enhanced intracellular dye accumulation (24, 27), rather than because of documented drug potentiating activity. However, the latter is not necessarily associated with observations of increased dye accumulation in the bacterial cell. An explanation might be the different time spans of dye efflux and accumulation vs susceptibility assays (200 s and 30 min, vs 20 h, respectively). In addition, differing physicochemical properties of dyes and antibiotic agents typically used in the respective test systems could be a reason (25, 27). Some substrate-specific activity had already been shown for the model EPIs PAβN (phenylalanine arginine β-naphthylamide) (32) and NMP (33).

### AcrB mutations with impact on the inhibitory potency of EPIs

In an earlier study using an *in vitro* random mutagenesis approach, we discovered AcrB double-mutation G141D_N282Y, which impaired the efflux inhibitory action of NMP in a predominantly substrate-specific manner. The two identified residues are located at the outside edge of the DBP and were critical for the activity of NMP with linezolid suggesting binding in or adjacent to this region (52). A similar impact of G141D_N282Y on the drug-enhancing potency of other EPIs could be expected if they share an overlapping binding site with that of NMP. Based on these considerations, we tested the selection of confirmed potent drug sensitizers with the AcrB double-mutant in combinational susceptibility assays. NMP and PAβN served as references, and we included a mutant harboring the AcrB substitution V411A, which was recently shown to eliminate the allosteric efflux inhibitory action of the pyridylpiperazine EPI BDM88855 (47). In contrast to G141_N282, residue V411 is located in the transmembrane region close to D407 and D408, which are known to be involved in the energy generating proton translocation process (54). However, efflux demonstrably remained unaffected in a V411A AcrB variant (47). Such a mutant derived from our AcrB overexpressing *E. coli* strain served as a control, because EPI activities impaired by G141D_N282Y should not be affected by V411A, and vice versa.

Among the EPIs with confirmed activity, the pyranopyridine compounds (MBX series) revealed an outstanding position regarding the impact of the G141D_N282Y

**TABLE 1** Drug sensitizing activity of EPIs in wild-type AcrB overexpressing *E. coli* 3-AG100

| EPI[b] | MIC EPI[c] (µg/mL) | EPI concn[d] [µg/mL (µM)] | Fold MIC decreases[a] | | | | | | | | | |
|---|---|---|---|---|---|---|---|---|---|---|---|---|
| | | | LVX | MXF | LZD | CLI | OXA | CXM | NOV | MIN | RIX | AZM |
| PAβN (16, 32) | >256 | 25 (48) | 4 | 9 | 12 | 16 | 7 | ≤2 | 137 | 10 | 69 | 21 |
| NMP (33) | >256 | 100 (440) | 8 | 7 | 31 | 12 | 6 | 4 | 6 | 9 | ≤2 | 5 |
| Mefloquine (24) | 80 | 18.9 (50) | ≤2 | ≤2 | ≤2 | ≤2 | ≤2 | ≤2 | ≤2 | ≤2 | ≤2 | ≤2 |
| Artesunate (26, 30) | >512 | 256 (670) | ≤2 | ≤2 | ≤2 | ≤2 | ≤2 | ≤2 | ≤2 | ≤2 | ≤2 | 3 |
| Sertraline (25) | 128 | 34.3 (100) | 4 | ≤2 | 4 | 12 | 4 | 4 | ≤2 | 4 | ≤2 | ≤2 |
| Pimozide (27) | >256 | 92.3 (200) | ≤2 | ≤2 | ≤2 | ≤2 | ≤2 | ≤2 | ≤2 | ≤2 | ≤2 | ≤2 |
| MBX2319 (35) | 64 | 9.7 (25) | 7 | 5 | 3 | 8 | 4 | 7 | ≤2 | 4 | ≤2 | 4 |
| Lanatoside C (17) | >256 | 128 (130) | ≤2 | ≤2 | ≤2 | ≤2 | ≤2 | ≤2 | ≤2 | ≤2 | ≤2 | ≤2 |
| Daidzein (17) | >256 | 128 (503) | ≤2 | ≤2 | ≤2 | ≤2 | ≤2 | ≤2 | ≤2 | ≤2 | ≤2 | ≤2 |
| NDGA (18) | >256 | 100 (330) | ≤2 | ≤2 | ≤2 | 8 | 4 | 4 | 3 | ≤2 | ≤2 | 15 |
| Mangiferin (18) | >256 | 100 (240) | ≤2 | ≤2 | ≤2 | ≤2 | ≤2 | ≤2 | ≤2 | ≤2 | ≤2 | ≤2 |
| Plumbagin- (18) | 128 | 32 (170) | ≤2 | ≤2 | ≤2 | ≤2 | ≤2 | ≤2 | ≤2 | ≤2 | ≤2 | ≤2 |
| Shikonin (18) | >256 | 100 (350) | ≤2 | ≤2 | ≤2 | ≤2 | ≤2 | ≤2 | ≤2 | ≤2 | ≤2 | ≤2 |
| Quercetin (18) | >512 | 256 (400) | ≤2 | ≤2 | ≤2 | ≤2 | ≤2 | ≤2 | ≤2 | ≤2 | ≤2 | ≤2 |
| MBX2931 (36, 50) | 128 | 13.2 (25) | 17 | 11 | 8 | 6 | 11 | 13 | 5 | 7 | 3 | 19 |
| MBX3132 (36, 50) | 128 | 6.2 (12.5) | 23 | 25 | 28 | 30 | 38 | 60 | 37 | 20 | 6 | 46 |
| MBX3135 (36, 50) | 128 | 6.3 (12.5) | 23 | 27 | 25 | 38 | 43 | 53 | 43 | 15 | 6 | 37 |
| MBX3796 (37) | 128 | 7.3 (12.5) | 17 | 17 | 23 | 29 | 32 | 40 | 27 | 16 | 5 | 53 |
| BM-19 (38) | 128 | 18.4 (50) | 3 | 5 | 7 | 4 | ≤2 | ≤2 | 6 | 4 | 8 | ≤2 |
| Procyanidin A2 (19) | >256 | 125 (220) | ≤2 | ≤2 | ≤2 | ≤2 | ≤2 | ≤2 | ≤2 | ≤2 | ≤2 | ≤2 |
| Reserpine (20) | 256 | 128 (200) | ≤2 | ≤2 | ≤2 | ≤2 | ≤2 | ≤2 | ≤2 | ≤2 | ≤2 | ≤2 |
| Domperidone (28) | >256 | 32 (80) | ≤2 | ≤2 | ≤2 | ≤2 | ≤2 | ≤2 | ≤2 | ≤2 | ≤2 | ≤2 |
| trans-Cinnamaldehyde (22) | 256 | 64 (480) | ≤2 | ≤2 | ≤2 | ≤2 | ≤2 | ≤2 | ≤2 | ≤2 | ≤2 | ≤2 |
| Amitriptyline (29) | >256 | 125 (460) | ≤2 | ≤2 | ≤2 | ≤2 | ≤2 | ≤2 | ≤2 | ≤2 | ≤2 | ≤2 |
| Chlorpromazine (29) | 256 | 64 (200) | ≤2 | ≤2 | ≤2 | ≤2 | ≤2 | ≤2 | ≤2 | ≤2 | ≤2 | ≤2 |
| Dihydrocapsaicin (21) | >256 | 100 (330) | ≤2 | ≤2 | ≤2 | ≤2 | ≤2 | ≤2 | ≤2 | ≤2 | ≤2 | 3 |
| 1-Benzyl-1,4-diazepane (43) | >256 | 100 (530) | ≤2 | ≤2 | 4 | ≤2 | ≤2 | ≤2 | ≤2 | ≤2 | ≤2 | 3 |
| NA #15 (45) | 40.6[e] | 20.3 (50) | 5 | 7 | 4 | 4 | 3 | 25 | 65 | ≤2 | ≤2 | 5 |
| NA #17 (45) | >71.5[e] | 17.9 (50) | 5 | ≤2 | 9 | 4 | ≤2 | ≤2 | ≤2 | ≤2 | 17 | 43 |
| NA #18 (45) | >71.5[e] | 8.9 (25) | ≤2 | ≤2 | 7 | ≤2 | ≤2 | ≤2 | ≤2 | ≤2 | 8 | 21 |
| NA #20 (45) | 65.5[e] | 16.4 (50) | 7 | 6 | 11 | 6 | 28 | 28 | 64 | 3 | 24 | 18 |
| NA #21 (45) | 73.5[e] | 18.4 (50) | ≤2 | ≤2 | 4 | ≤2 | ≤2 | ≤2 | ≤2 | ≤2 | ≤2 | ≤2 |
| BDM88855 (47) | >256 | 28.4 (100) | 12 | 16 | 32 | 16 | 21 | 32 | 16 | 16 | 4 | 32 |
| PCPP (46) | >256 | 100 (509) | 9 | 6 | 6 | 4 | 4 | ≤2 | 4 | 6 | ≤2 | 8 |
| RP1 (23) | >256 | 128 (587) | ≤2 | ≤2 | ≤2 | ≤2 | ≤2 | ≤2 | ≤2 | ≤2 | ≤2 | ≤2 |
| Proguanil (31) | >256 | 100 (340) | 6 | 12 | 14 | 11 | 8 | 6 | 11 | 11 | 4 | 16 |
| Tafenoquine (31) | 64 | 16 (30) | 3 | ≤2 | 4 | 5 | 4 | 6 | ≤2 | 3 | ≤2 | ≤2 |

*(Continued on next page)*

TABLE 1 Drug sensitizing activity of EPIs in wild-type AcrB overexpressing *E. coli* 3-AG100 (*Continued*)

| EPI[b] | MIC EPI[c] (µg/mL) | EPI concn[d] [µg/mL (µM)] | Fold MIC decreases[a] | | | | | | | | | | |
|---|---|---|---|---|---|---|---|---|---|---|---|---|---|
| | | | LVX | MXF | LZD | CLI | OXA | CXM | NOV | MIN | RIX | AZM |
| EDHB (49) | >256 | 128 (703) | ≤2 | ≤2 | ≤2 | ≤2 | ≤2 | ≤2 | ≤2 | ≤2 | ≤2 | ≤2 |

[a]Fold MIC decrease with EPI (ratio of MICs without and with EPI, decreases ≥4-fold given bold-faced); MIC of the drug alone given in Table S1. LVX, levofloxacin; MXF, moxifloxacin; LZD, linezolid; CLI, clindamycin; OXA, oxacillin; CXM, cefuroxime; NOV, novobiocin; MIN, minocycline; RIX, rifaximin; AZM, azithromycin.

[b]References given in parentheses. NMP, 1-(1-naphthylmethyl)piperazine; NDGA, nordihydroguaretic acid; PCPP, p-chlorophenylpiperazine; NA, naphthylamide EPI; EDHB, ethyl 3,4-dihydroxybenzoate.

[c]MICs determined with the AcrB overexpressing *E. coli* 3-AG100, if not otherwise indicated.

[d]Concn, concentration. EPI concs used in the combinational MIC assays correspond to those published. In the case of lacking activity with the combined drugs, higher doses were applied (maximum at subinhibitory concn).

[e]MIC from the indicated publication (determined with wt-AcrB *E. coli* BW25113).

double-mutation. Remarkably decreased sensitizing with drugs from nearly all antibiotic classes was detected. Even with the less potent inhibitors MBX2319 and MBX2931, further activity loss was observed (Fig. 1 and 2). This finding somewhat contradicts the hypothesis that the highly linezolid-specific impact of G141D_N282Y on the activity of NMP is due to the differential action of this EPI, which is most effective with linezolid (55). The activities of the MBX EPIs were also significantly impaired when used with the more lipophilic and larger drugs novobiocin and azithromycin. Only subtle changes due to G141D_N282Y were detected with rifaximin, which is an extremely lipophilic and high-molecular weight drug (Fig. 2). However, the potentiating activity even of MBX3132 and MBX3135 with this antibiotic was also low in the wild-type AcrB strain (Table 1). Despite its large size, rifaximin is a confirmed efflux substrate demonstrated by 8–16-fold increased susceptibility of AcrB deficient *E. coli* mutants (2). It belongs to the rifamycins, which are supposed to bind in the PBP of AcrB (12). The susceptibility to rifaximin was found most significantly enhanced with PAβN. The increased activity of this EPI with high-molecular weight drugs is at least in part explainable by an OM permeabilizing activity (32, 53) contributing to efflux inhibition and presumably facilitating the passage of large molecules through the Gram-negative cell envelope. A relatively low potentiating activity in combination with rifaximin might indicate in turn less or no impact on the OM integrity confirming previous results with the MBX agents (36). Besides PAβN, the naphthylamide compounds #17 and #20 showed sensitizing with rifaximin suggesting possible membrane effects in addition to efflux inhibition (Fig. 2).

In contrast to the MBX compounds and NMP, the drug potentiating activity of the EPIs BM-19, nordihydroguaretic acid (except with azithromycin), proguanil, tafenoquine, and BDM88855 was not reduced when tested in mutant G141D_N282Y. Minor effects were detectable with p-chlorophenylpiperazine, and very substrate-specific differential decreases were observed in the potency of the naphthylamide EPIs NA #17 and NA #20 (Fig. 1 and 2).

Corresponding to recently published data (47), we confirmed an overall loss in activity of the broad-effective EPI BDM88855 in the V411A AcrB variant. A new and surprising finding was that in this mutant, the MBX, but not the other EPIs, showed increased potentiating with almost all of the tested drugs except levofloxacin, minocycline, and rifaximin (Fig. 1 and 2). Because of these observations and in addition to the fact that even AcrB deficiency did not increase the susceptibility to the MBX compounds (data not shown), an intrinsic antibacterial action of these EPIs in the V411A mutant can be excluded. So far, it is not clear, how this mutation in the AcrB transmembrane region is able to enhance the efflux inhibitory action of the MBX compounds. There could exist a transport route via the transmembrane region to the site of action for these EPIs in the DBP, but an allosteric effect of the V411A mutation might also be conceivable.

## Impact of MBX3135 and BDM88855 on the intracellular accumulation of linezolid

As demonstrated above, the double-mutation G141D_N282Y was associated with substantial resistance to the drug potentiating activity of the MBX EPIs. To investigate this phenomenon further, we carried out linezolid uptake assays in the absence and presence of MBX3135, which is one of the most potent MBX compounds. In addition, we tested BDM88855 as a non-DBP binding EPI (47). With MBX3135, we found significantly decreased intracellular concentrations of linezolid in mutant G141D_N282Y compared to the parental *E. coli* strain confirming that the mutated efflux pump was inhibited much less effectively than the wild-type transporter. In contrast, the difference between the two strains remained negligible when using the BDM88855 inhibitor (Fig. 3). With the intracellular uptake measurement, we corroborate the findings from susceptibility testing by excluding long-term effects possibly playing a role in the latter. Synergy changes of MBX3135 with linezolid in MIC as well as in intracellular accumulation assays are similar to those previously demonstrated for NMP in mutant G141D_N282Y (52).

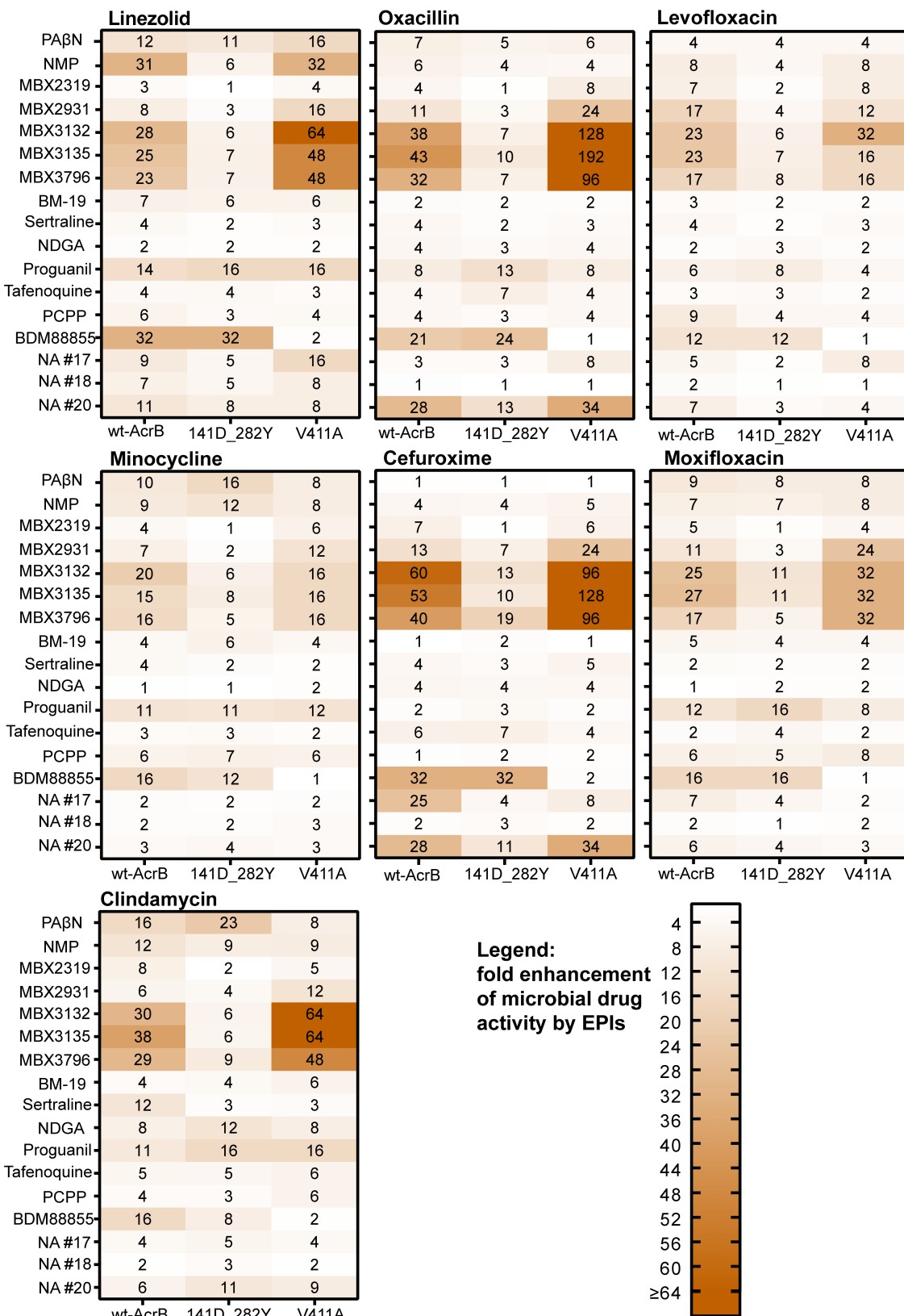

**FIG 1** Potentiating activity of EPIs with drugs of lower molecular weight (<500) and lipophilicity determined in wt-AcrB *E. coli* 3-AG100 and AcrB mutants G141D_N282Y and V411A (ratios of the MICs without and with EPI shown, *n* ≥ 3). Used EPI concentrations as indicated in Table 1 except sertraline and nordihydroguaretic acid (NDGA), that were used at 50 µM and 165 µM with G141D_N282, respectively (increased intrinsic susceptibility of the mutant to these compounds).

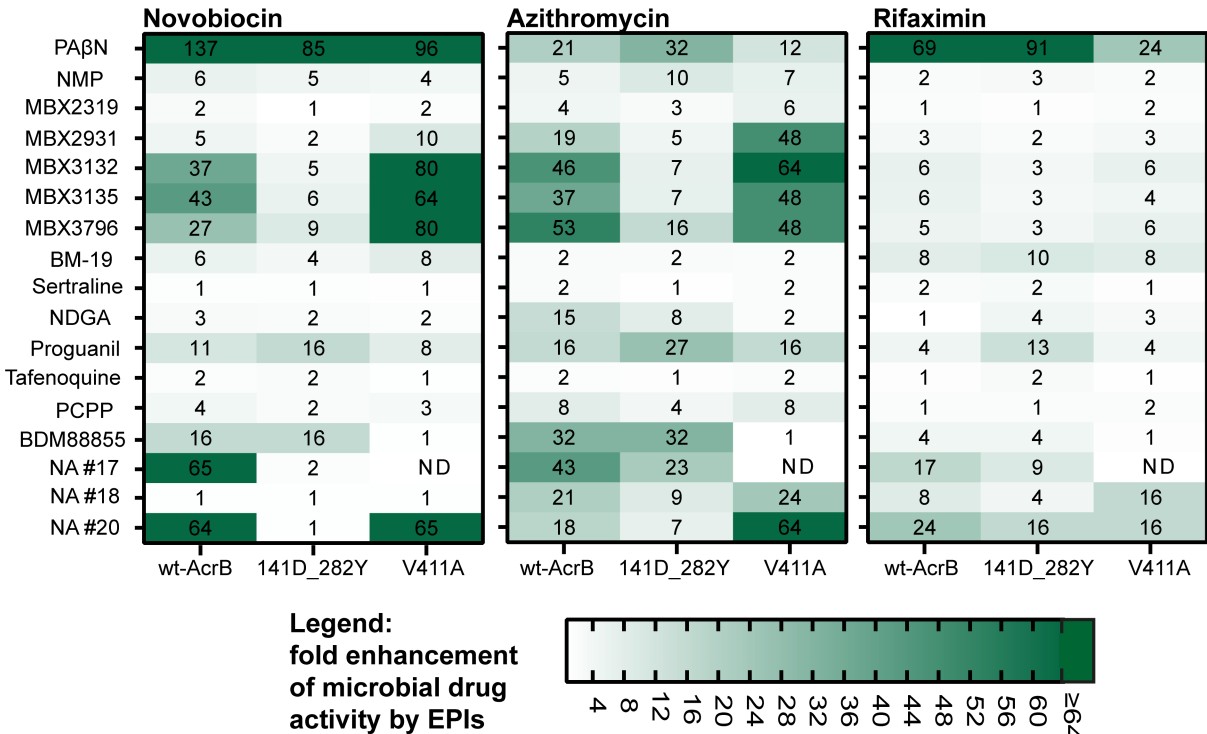

Legend: fold enhancement of microbial drug activity by EPIs

4 8 12 16 20 24 28 32 36 40 44 48 52 56 60 ≥64

**FIG 2** Potentiating activity of EPIs with drugs of higher molecular weight (>600) and lipophilicity determined in wt-AcrB *E. coli* 3-AG100 and AcrB mutants G141D_N282Y and V411A (ratios of the MICs without and with EPI shown, $n \geq 3$). Used EPI concentrations as indicated in Table 1 except sertraline and NDGA, that were used at 50 µM and 165 µM with G141D_N282, respectively (increased intrinsic susceptibility of the mutant to these compounds).

## G141_N282 during conformational cycling in RND transporter structures

To approach the possible role of the G141_N282 site (Fig. 4) for the action of EPIs, it should be illuminating to examine the relative movement of these residues to one another during conformational cycling (from the access to the binding and to the extrusion state). For this purpose, we measured distances between the two AcrB residues (or the corresponding homologs in other RND transporters, Tables S2 and S3) in chains A, B, and C of 61 asymmetric structures obtained from the RCSB Protein Data Bank

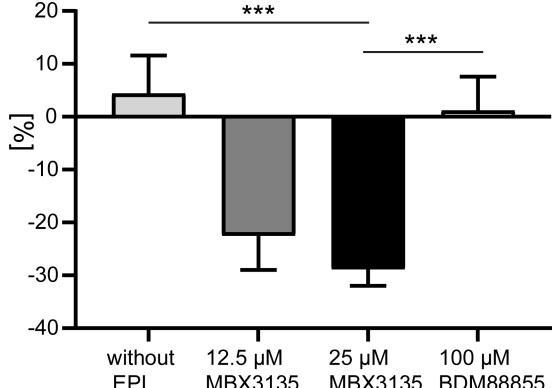

**FIG 3** Intracellular linezolid accumulation in AcrB mutant G141D_N282Y relative to that in wt-AcrB strain 3-AG100 (mean values with standard error of the mean after 20 min of incubation); statistical significance in differences was indicated (***$P \leq 0.001$; sample sizes: $n = 12$ without EPI and with BDM88855, $n = 2$ with 12.5 µM MBX3135, $n = 10$ with 25 MBX3135). Negative values indicate less intracellular accumulated drug in G141D_N282Y compared to 3-AG100.

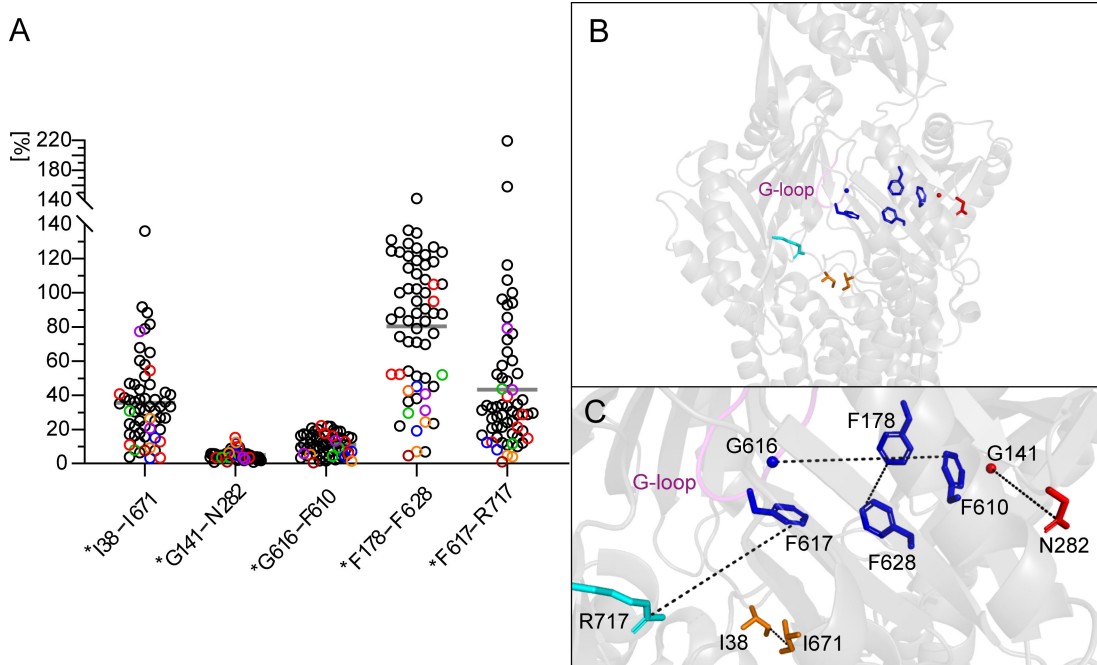

**FIG 4** (A) Distribution of maximum distance changes between residues of amino acid pairs in AcrB and homologous RND transporters during the conformational transition from the access, to the binding and to the extrusion state (Tables S2 and S3). Black circles, AcrB; red, MexB; green, CmeB; blue, MtrD; orange, AdeB; purple, AdeJ; and brown, OqxB. (B) Position of residue pairs used for measurements in AcrB [PDB structure 7OUK chain B (47)]. Amino acid side chains are depicted as sticks, glycine as spheres; blue, distal binding pocket residues; red, residues critical for NMP activity; cyan, proximal binding pocket residue; orange, residues of a putative transport route for low-molecular weight drugs; and magenta, G-loop. (C) AcrB section, distance measurements are given as dotted lines.

(PDB) and determined changes during transition of the protomers (Fig. 4; Table S2). In order to put the results in a relation, we also examined distance changes between residues (i) G616 (DBP, G-loop) and F610 (DBP), (ii) F178 (DBP) and F628 (DBP), (iii) F617 (DBP, G-loop) and R717 (PBP, cleft-side), and (iv) I38 and I671 [both belonging to a substrate-selective putative entrance channel (56)] (Fig. 4). G616/F610 and F178/F628 roughly represent a horizontal and vertical crossing of the DBP (Fig. 4). All of the residues selected for measurements, except D141 and N282, have been shown to contribute to substrate selectivity (11, 56–58). Largest average changes during conformational transition were found between DBP residues F178 and F628 (80%), F617 and the PBP residue R717 (43%), and the putative entrance channel residues I38 and I671 (36%) (Fig. 4; Table S2). Significant movements of residues lining the substrate routes could be expected. The relative inflexibility of DBP residues F610 and G616 reflected by an average distance change of 11% appeared surprising, particularly because the latter belongs to the G-loop, which has been described as a flexible element between the PBP and DBP (13). It should be noted that of course, the scattering of values is partly due to differing methods used to generate protein crystals (crystallization conditions) and structures (X-ray, cryo-EM). In this context, it is particularly remarkable that the change in distance between N282 and G141 was only 5% on average with minor deviations (Fig. 4). The spatial arrangement of the two amino acids appeared almost static during conformational cycling not only in AcrB but also in homologous transporters from other species and independently if ligands were cocrystallized or not (Fig. 4; Table S2). Notably, G141 and N282, in particular the latter, are highly conserved among RND transporters (Table S3) suggesting an essential role in proper pump functioning.

## Possible role of G141D_N282Y for EPI activity

It could be speculated that MBX EPIs act by binding close to the G141_N282 site and thereby perturbing its spatial arrangement, which appeared conserved during

conformational cycling. In double-mutant G141D_N282Y, an additional stabilization might occur by an H-bond between asparagine and tyrosine possibly preventing a disarrangement and by that diminishing the activity of the MBX compounds and NMP. This hypothesis was supported by previous results showing that single mutations G141D and N282Y were not able to impair the NMP potency (52). Alternatively, an effect of EPIs on F610 was conceivable, because it is, among the DBP phenylalanines, located closest to the G141_N282 site (52) (Fig. 4). Furthermore, cocrystallization of fractional AcrB with MBX3132 had shown an overlapping binding of the EPI with the substrates minocycline, rhodamine 6G, and doxorubicin, but closer to F610 (50). Altered polarity and bulkiness of the substituted residues in G141D_N282Y might affect the binding of MBX compounds and NMP next to F610. A critical role of the latter for the resistance to a broad range of drugs and for dye efflux had been demonstrated in mutagenesis studies, whereas the other DBP phenylalanines had shown less or more substrate-specific impact (11, 59). Notably, despite the supposed structural importance of G141_N282, minor effects on the resistance to only a few of the tested drugs were observed in the double-mutant (Table S1), and unaffected dye efflux had previously been proven (52).

## Summary and conclusions

To the best of our knowledge, this has been the most comprehensive comparative study of EPI activity in *E. coli* so far. Under the conditions used here (test strain with proven AcrB overexpression, range of confirmed AcrB efflux substrates from several antibiotic classes, fourfold growth inhibitory action for at least two drugs), we observed quite variable *in vitro* potency of the evaluated EPIs with some of the agents showing no relevant activity. Pyranopyridine derivatives (MBX compounds) were among the most active EPIs. They were largely affected by the AcrB double-mutation G141D_N282Y, previously only known to impair the inhibitory action of NMP and here shown to leave the activity of other EPIs including BDM88855 unchanged. These findings do not only suggest the need for more comparative EPI research with well-defined conditions. They also demonstrate the usefulness of selected efflux pump variants potentially allowing an initial classification of EPIs with regard to their mechanism of function and providing indications of the structural requirements for effective inhibition of RND transporters.

## MATERIALS AND METHODS

### Strains, growth conditions, and chemicals

*E. coli* strains and mutants used in the present study are listed in Table 2. Bacteria were cultivated at 37°C on LB (Luria/Miller broth) agar plates overnight or in liquid LB medium (Roth, Karlsruhe, Germany) as indicated. Chemicals were purchased from Sigma-Aldrich (Taufkirchen, Germany) with the exception of NMP, that was from Chess (Mannheim, Germany), chlorpromazine, mangiferin, shikonin, procyanidin A2, sertraline, and pimozid, that were from Hycultec (Beutelsbach, Germany). The MBX compounds were a kind gift from Thimothy J. Opperman (Microbiotix, Worcester, MA, USA), BM-19 from Jadwiga Handzlik, the substituted naphthylamides from the group of John K. Walker (Department of Pharmacology and Physiology, Saint Louis University School of Medicine, St. Louis,

**TABLE 2** Strains and mutants used in this study

| *E. coli* strains/mutants | Description | Source |
|---|---|---|
| 3-AG100 | AG100 derivative (wild-type *acrB* overexpressing *E. coli* strain) | Kern et al. (8) |
| AcrB mutants[a,b] | | |
| G141D_N282Y | AcrB mutant derived from parental strain 3-AG100 | Schuster et al. (52) |
| V411A | AcrB mutant derived from parental strain 3-AG100 | This study |

[a]Generated by site-directed chromosomal recombination.
[b]Similar expression of the double-mutation containing and the wild-type AcrB protein had been demonstrated in western blot experiments from Jean-Marie Pagès and Jean-Michel Bolla (unpublished data). Gene expression data of both mutants are provided in the Supplemental Material (Fig. S1).

MO, USA), and BDM88855 from Marion Flipo (Univ. Lille, Inserm, Institut Pasteur de Lille, U1177-Drugs and Molecules for Living Systems, Lille, France).

## Drug-enhancing potency and MIC testing

The drug-enhancing potency of EPIs was examined in standard MIC assays (performed with LB broth) with and without EPIs as described previously (52) using 96-well custom plates prefilled with lyophilized drugs (Merlin, Bornheim-Hersel, Germany). Growth controls in the absence and presence of the respective EPI concentrations (without drug) were carried out in parallel. For the MIC testing of EPIs, twofold dilutions of the compounds were prepared and inoculated in the same way as for the standard MIC assays (52).

## Linezolid uptake assay

The intracellular accumulation of linezolid was determined according to a method published earlier (60). Briefly, LB medium was inoculated with an overnight grown culture to an optical density ($OD_{600}$)of 0.02. The bacteria were cultivated at 37°C with shaking to an $OD_{600}$ of 0.5–0.6 and then harvested by centrifugation at 3,320 × $g$ (10 min). After washing twice with phosphate buffered saline (PBS), the cells were suspended in PBS containing 0.4% D-glucose and adjusted to an $OD_{600}$ of 0.4. For drug uptake in the presence of EPI, the bacterial suspension was supplemented with the compound. After 10 min at 37°C with shaking, linezolid was added to a final concentration of 20 µg/mL and the incubation continued. An aliquot of 1 mL was taken every 10 min and immediately centrifuged (16,000 × $g$, 4°C) through a silicon oil double layer [335 µL AR200, 165 µL AK100 (Wacker, Burghausen, Germany)] to separate the cells from the drug-containing buffer. Pellets were lysed using 500 µL 0.1 M glycine-HCl buffer, pH 3, at room temperature overnight. After 10 min of centrifugation at 16,000 × $g$, the supernatants were collected and stored at −20°C for linezolid measurement.

## Linezolid determination with HPLC-MS

Linezolid was measured using high-performance liquid chromatography with combined mass spectrometry (HPLC-MS). Samples were prepared by mixing 100 µL of the supernatant from the cell lysis with 500 µL of an internal standard containing 1 µg/mL oxacillin (in $H_2O$). Then, 10 µL aliquots were injected into the HPLC-MS device (details of instruments, run conditions, and MS parameters for linezolid and oxacillin, see Table S4). Concentrations were calculated using the internal standard and external calibration curves of linezolid (0–1.2 µg/mL, 9 measurement points, $R^2 > 0.95$).

## Chromosomal site-directed mutagenesis

The AcrB V411A *E. coli* mutant was constructed by chromosomal substitution using homologous recombination procedures as described previously. In the first step, an *acrB* mutant was generated from parental strain 3-AG100, in which the nucleotides encoding AcrB amino acids 403–413 were replaced by an *rpsLneo*-cassette (55). In the second step, *rpsLneo* in *acrB* was substituted by a repair oligonucleotide with the following sequence (synthesized from Micro-synth, Balgach, Swiss): 5′-*acacgctaacaatgttcgggatggtgctcgccatcggcctgttggtggatgacgc-catcgccgtggtagaaaacgttgagcgtgttatggcggaagaaggttt*-3′ (underlined triplet encoding V411A). Successfully recombined clones were selected using LB agar supplemented with 16 µg/mL linezolid (MIC with the *acrB::rpsLneo* mutant was 8 µg/mL), and Sanger sequencing was performed to confirm the substitution.

## Structure visualization and distance measurements

RND efflux transporter structures were from the RCSB PDB (https://www.rcsb.org/). Images and distance measurements were created using PyMOL (references and details are given in the Supplemental Material, Tables S2 and S3). Distances between indicated residues were measured in chains A, B, and C (using defined atoms, Table S3) of asymmetric structures, and the maximum (Δmax) and minimum distances (Δmin) were determined. The maximum percent distance change was calculated as

$$\% \max \text{dist.change} = (\Delta\max - \Delta\min) \times \frac{100}{\Delta\min}$$

## Statistical analysis

Statistical significance of differences was analyzed by unpaired two-tailed $t$-tests using the GraphPad Prism software version 9.5.0 (San Diego, CA, USA).

## ACKNOWLEDGMENTS

We thank the following scientists and institutions for kindly providing EPI compounds: Timothy J. Opperman (Microbiotix, Worcester, Massachusetts, USA) for the pyrano-pyridine inhibitors MBX2319, MBX2931, MBX3132, MBX3135, and MBX3796; John K. Walker (Department of Pharmacology and Physiology, Saint Louis University School of Medicine, St. Louis, Missouri, USA) for the substituted naphthylamides; Marion Flipo (Univ. Lille, Inserm, Institut Pasteur de Lille, U1177-Drugs and Molecules for Living Systems, Lille, France) for the pyridylpiperazine compound BDM88855; Jadwiga Handzlik for the piperazine arylideneimidazolone BM-19. We also thank Jean-Marie Pagès and Jean-Michel Bolla (Université Aix-Marseille, 13385 Marseille, France) for including mutant G141D_N282Y and wild-type AcrB strain 3-AG100 in western blot analysis.

Finally, we acknowledge support by the Open Access Publication Fund of the University of Freiburg.

## AUTHOR AFFILIATIONS

[1]Division of Infectious Diseases, Department of Medicine II, University Hospital and Medical Center, Freiburg, Germany
[2]Faculty of Medicine, Albert-Ludwigs University, Freiburg, Germany

## AUTHOR ORCIDs

Sabine Schuster  http://orcid.org/0000-0002-8817-6224

## FUNDING

| Funder | Grant(s) | Author(s) |
|---|---|---|
| Open Access Publication Fund of the University of Freiburg | | Winfried V. Kern |

## AUTHOR CONTRIBUTIONS

Sabine Schuster, Conceptualization, Investigation, Methodology, Validation, Visualization, Writing – original draft | Martina Vavra, Investigation | Dave A. N. Wirth, Investigation, Methodology | Winfried V. Kern, Conceptualization, Project administration, Supervision, Writing – review and editing

## ADDITIONAL FILES

The following material is available online.

## Supplemental Material

**Table S1 to S4, Fig. S1 and S2 (Spectrum03045-S0001.pdf).** Tables S1 (MICs of drugs alone and fold-MIC decreases with EPIs), S2 (Conformational distance changes in AcrB), S3 (Atoms used for distance measurements), and S4 (HPLC_MS parameters for linezolid measurement) and Fig. S1 (*acrB* gene expression data) and S2 (Growth curves of strains).

## Open Peer Review

**PEER REVIEW HISTORY (review-history.pdf).** An accounting of the reviewer comments and feedback.

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
