## [Reviewer comments · Microbiology Spectrum]

Microbiology Spectrum

Comparative reassessment of AcrB efflux inhibitors reveals differential impact of specific pump mutations on the activity of potent compounds

Sabine Schuster, Martina Vavra, Dave Wirth, and Winfried Kern

Corresponding Author(s): Sabine Schuster, University Hospital and Medical Center Freiburg

Review Timeline:

Submission Date:	August 8, 2023
Editorial Decision:	October 9, 2023
Revision Received:	November 14, 2023
Accepted:	November 17, 2023

Editor: Monika Kumaraswamy

Reviewer(s): Disclosure of reviewer identity is with reference to reviewer comments included in decision letter(s). The following individuals involved in review of your submission have agreed to reveal their identity: Xian-Zhi Li (Reviewer #3)

Transaction Report:

DOI: <https://doi.org/10.1128/spectrum.03045-23>

October 9, 2023

Dr. Sabine Schuster
University Hospital and Medical Center Freiburg
Division of Infectious Diseases, Medicine II
Hugstetter Str. 55
Freiburg 79106
Germany

Re: Spectrum03045-23 (Comparative reassessment of AcrB efflux inhibitors reveals differential impact of specific pump mutations on the efficacy of potent compounds)

Dear Dr. Sabine Schuster:

Thank you for submitting your manuscript to Microbiology Spectrum. Following careful evaluation of your manuscript on the comparative reassessment of AcrB efflux inhibitors, the Reviewers have determined that your study is of significant interest and importance. We would be pleased to consider a revised version provided that you can satisfactorily address the substantive concerns raised in the reviews (appended below).

Link Not Available

Sincerely,

Monika Kumaraswamy, MD, D(ABMM)
Editor
Microbiology Spectrum

Journals Department
Reviewer comments:

Reviewer #1 (Comments for the Author):

Please see attached.

Reviewer #2 (Comments for the Author):

This ms reports comparative reassessment of the reported drug efflux pump inhibitors (EPIs) for their activity against the prototype RND transporter, AcrB of *E. coli*. Impacts from AcrB mutations on EPI activity were observed. First, EPIs of drug resistance pumps have been widely reported in literature, though none of them are used clinically. However, a significant portion of these studies can be highly questionable regarding their findings, e.g., whether the reported compounds display true efflux inhibition? Were there effects on membrane permeability or other non-efflux process. This current ms has provided a parallel comparison of a large number of so-called EPIs, which is much in need, and thus, this ms is of high biomedical importance for (positively) provocative questions on the reported EPIs. This investigation is carefully designed, conducted and validated using AcrB-defined isogenic strains and biochemical approach. This ms is well articulated for its rationale, finding and discussion. The findings shed light on the EPIs of RND pumps, including effects from the transporter mutation. The reviewer merely has some minor editorial comments as detailed below.

L2/L30/L40/L120/L150: "Efficacy" often means in vivo effects. But this study is in vitro, strongly suggest changing "efficacy" to "activity" as used in L22 (Abstract) or to "potency" seen in L78.

L18/L57. Spell out "E. coli" for its first appearance.

L31. The first "providing" can be deleted or be replaced by another word to avoid two "providing" in the same sentence.

L44. Spell out "NMP" in Importance, an independent section (but remove "NMP" in L28 in Abstract).

L51 Would write "resistance-nodulation-cell division" with dashes per most literature. For this first sentence, would use "Efflux by the resistance-nodulation-cell division (RND) superfamily transporters", rather than "type" to have a higher attitude.

L58. Change "proteins" before "AcrA" to "protein" (singular).

L59/L62/L67/the ms. "RND-type" is used throughout the ms. To be somehow user friendly, the authors may also use "RND" such as "RND transporters" or "RND pumps" without "type".

L67. "RND-type efflux" may better be changed to "RND pumps".

L87-88. What is "RND-type efflux inhibition"? Suggest rewording.

L157. Would add "of AcrB" after "PBP" to have "PBP of AcrB".

L255. Write "Table 2".

Staff Comments:

Preparing Revision Guidelines

Please return the manuscript within 60 days; if you cannot complete the modification within this time period, please contact me. If you do not wish to modify the manuscript and prefer to submit it to another journal, please notify me of your decision immediately so that the manuscript may be formally withdrawn from consideration by Microbiology Spectrum.

The article by Schuster et al provides a comprehensive analysis of the impact of different efflux pump inhibitors on drug resistant *E. coli* strains. The article is a very interesting read and is also a well written one. The results are supported by strong data. Here are my comments:

- 1) Is the efflux pump protein with double mutation expressed equally like the wildtype protein? Please clarify.
- 2) Are growth rates of the strain carrying double mutations are different compared to the wild-type, since growth rates can affect MICs? Please clarify.
- 3) Which media was used to perform the MICs? Please clarify.
- 4) Did the authors check the gene/protein expression of AcrB V411A *E* once it was constructed? Please clarify.
- 5) Fig 3: Is the intracellular accumulation by 12.5uM MBX significantly different from the accumulation without EPI? Please clarify.
- 6) Fig 3: Please clarify in the text what negative accumulation signifies?
- 7) Does any of the EPIs inhibit growth of *E.coli* in absence of drugs? Please clarify.

Response to reviewer comments

(Spectrum03045-23. Indicated line numbers refer to the revised manuscript):

Reviewer #1:

1) *Is the efflux pump protein with double mutation expressed equally like the wildtype protein?*

The mutant with the G141D_N282Y mutation and its parent had been included in western blot analysis performed by the lab of Jean-Marie Pagès and Jean-Michel Bolla in 2014. Similar expression of the wild type and the double mutated AcrB protein had been shown (unpublished data). The results were available, when the manuscript concerning the double mutant had already been submitted. We acknowledge the western blot analysis and give an indication in the legend of Table 2 in the revised manuscript.

2) *Are growth rates of the strain carrying double mutations are different compared to the wild-type, since growth rates can affect MICs?*

The mean doubling time in the exponential phase is 60 min with the double-mutant and 54 min with the wild-type AcrB strain 3-AG100 (58 min with the V411A AcrB mutant). We provide growth curves in LB broth in the revised supplement (Fig. S2). The MICs of all strains are shown in Table S1 (for drugs alone, rows highlighted in grey). In fact, the double-mutant was slightly more susceptible to many but not to all of the tested drugs. In view of this circumstance, a rather higher effectiveness of the EPIs was to be expected. Instead, the NMP and the MBX activity was substantially decreased, whereas the activity of the other EPIs were not more than marginally affected.

3) *Which media was used to perform the MICs?*

All assays in this study including MIC testing were carried out with LB broth as indicated in the method section "Strains, growth conditions, and chemicals" and by referring to reference #52 in the section "Drug enhancing potency and MIC testing". LB had been used in all previous studies with strains and mutants related to the *E. coli* AG100 series (see references #11, 15, 25, 27, 31, 52, 55-59). Different media that were used for EPI testing could be found in the literature. Our aim in the present study was to achieve comparability of the EPIs with each other under the identical conditions.

For clarity, we now have added “(performed with LB broth)” (L269/270) in the method section “Drug enhancing potency and MIC testing”.

4) *Did the authors check the gene/protein expression of AcrB V411A E once it was constructed?*

We checked the *acrB* expression in the AcrB mutants and added the results in the revised supplement (Fig. S1). Wild-type AcrB strain *E. coli* 3-AG100 and the mutants revealed similar gene expression. No differences in protein expression of AcrB containing V411A and wild-type AcrB had been shown in a previous publication studying the effect of the chromosomal mutation on the activity of pyridylpiperazine inhibitors in an *E. coli* BW25113 strain (reference #47).

5) *Fig 3: Is the intracellular accumulation by 12.5uM MBX significantly different from the accumulation without EPI?*

Statistical significance of the differences was found as indicated with asterisks (Fig. 3). It was not found for accumulation changes in the presence of 12.5 μ M MBX vs. without EPI.

Therefore, we additionally performed assays with 25 μ M MBX. Lower activity of 12.5 μ M in drug uptake assays compared to that in MIC assays might be due to the different time spans of these assays (20 min vs. 20 h). We have now specified the statistical significance more detailed in the legend of Fig. 3.

6) *Fig 3: Please clarify in the text what negative accumulation signifies.*

In Fig 3, we show the intracellular accumulated linezolid in the double-mutant relative to that in the wt-AcrB strain 3-AG100 due to EPI exposure (or without EPI). Negative values indicate less accumulation of linezolid in the double-mutant than in the wt-AcrB *E. coli* (as seen in the presence of the MBX EPI). We now replaced the first sentence of the legend by “Intracellular linezolid accumulation in AcrB mutant G141D_N282Y relative to that in wt-AcrB strain 3-AG100” and added “The negative values indicate less intracellular accumulated drug in G141D_N282Y compared to 3-AG100” in the legend of Fig 3.

7) *Does any of the EPIs inhibit growth of E.coli in absence of drugs?*

The MICs of the EPI compounds (without drug) are given in Table 1 in (second column) and the EPI concentrations used in combination with drug in the third column. The applied EPI concentrations in the combinational MIC assays did not exceed $\frac{1}{4}$ MIC (of the EPI). In

addition, growth controls with and without the respective EPI concentrations were always carried out in parallel with the assays. This was now additionally indicated in the method section "Drug enhancing potency and MIC testing" (L271-L272).

Reviewer #2:

- L2/L30/L40/L120/L150: *"Efficacy" often means in vivo effects. But this study is in vitro, strongly suggest changing "efficacy" to "activity" as used in L22 (Abstract) or to "potency" seen in L78.*

In the revised manuscript, "efficacy" has now been changed to "activity" or to "potency".

- L18/L57. *Spell out "E. coli" for its first appearance.*

As recommended, "E. coli" has been changed to "Escherichia coli" (in L18/L58).

- L31. *The first "providing" can be deleted or be replaced by another word to avoid two "providing" in the same sentence.*

First "providing" has been deleted (L31).

- L44. *Spell out "NMP" in Importance, an independent section (but remove "NMP" in L28 in Abstract).*

As recommended, we included "1-(1-naphthylmethyl)piperazine" in L44 and removed "NMP" in L28.

- L51 *Would write "resistance-nodulation-cell division" with dashes per most literature. For this first sentence, would use "Efflux by the resistance-nodulation-cell division (RND) superfamily transporters", rather than "type" to have a higher attitude.*

We followed the recommendation for L51 and wrote "Efflux by the resistance-nodulation-cell division (RND) superfamily transporters" (L52).

- L58. *Change "proteins" before "AcrA" to "protein" (singular).*

According to current knowledge, the AcrAB-TolC complex is stabilized by six AcrA proteins (L60).

- *L59/L62/L67/the ms. "RND-type" is used throughout the ms. To be somehow user friendly, the authors may also use "RND" such as "RND transporters" or "RND pumps" without "type".*
In most cases, "RND-type" has now been replaced by "RND" (in L244 by "AcrB").
- *L67. "RND-type efflux" may better be changed to "RND pumps".*
We changed "RND-type efflux" by "efflux by RND pumps" (L69).
- *L87-88. What is "RND-type efflux inhibition"? Suggest rewording.*
Rewording to "in AcrB regarding the inhibition of efflux" was done (L90).
- *L157. Would add "of AcrB" after "PBP" to have "PBP of AcrB".*
As recommended, "of AcrB" was added after "PBP" (L159).
- *L255. Write "Table 2"*
Correction done (L257).

Re: Spectrum03045-23R1 (Comparative reassessment of AcrB efflux inhibitors reveals differential impact of specific pump mutations on the activity of potent compounds)

Dear Dr. Sabine Schuster:

Your manuscript has been accepted, and I am forwarding it to the ASM production staff for publication. Your paper will first be checked to make sure all elements meet the technical requirements. ASM staff will contact you if anything needs to be revised before copyediting and production can begin. Otherwise, you will be notified when your proofs are ready to be viewed.

Sincerely,
Monika Kumaraswamy, MD, D(ABMM)
Editor
Microbiology Spectrum